# A CRITICAL LOOK AT THE EVALUATION OF GNNS UNDER HETEROPHILY: ARE WE REALLY MAKING PROGRESS?

**Oleg Platonov**
HSE University, Yandex Research
olegplatonov@yandex-team.ru

**Denis Kuznedelev**
Skoltech, Yandex Research
dkuznedelev@yandex-team.ru

**Michael Diskin**
HSE University, Deepcake.io
michael.s.diskin@gmail.com

**Artem Babenko**
Yandex Research
artem.babenko@phystech.edu

**Liudmila Prokhorenkova**
Yandex Research
ostroumova-la@yandex-team.ru

## ABSTRACT

Node classification is a classical graph representation learning task on which Graph Neural Networks (GNNs) have recently achieved strong results. However, it is often believed that standard GNNs only work well for *homophilous* graphs, i.e., graphs where edges tend to connect nodes of the same class. Graphs without this property are called *heterophilous*, and it is typically assumed that specialized methods are required to achieve strong performance on such graphs. In this work, we challenge this assumption. First, we show that the standard datasets used for evaluating heterophily-specific models have serious drawbacks, making results obtained by using them unreliable. The most significant of these drawbacks is the presence of a large number of duplicate nodes in the datasets squirrel and chameleon, which leads to train-test data leakage. We show that removing duplicate nodes strongly affects GNN performance on these datasets. Then, we propose a set of heterophilous graphs of varying properties that we believe can serve as a better benchmark for evaluating the performance of GNNs under heterophily. We show that standard GNNs achieve strong results on these heterophilous graphs, almost always outperforming specialized models. Our datasets and the code for reproducing our experiments are available at https://github.com/yandex-research/heterophilous-graphs.

## 1 INTRODUCTION

The field of machine learning on graph-structured data has recently attracted a lot of attention, with Graph Neural Networks (GNNs) achieving particularly strong results on most graph tasks. Thus, using GNNs has become a de-facto standard approach to graph machine learning, and many versions of GNNs have been proposed in the literature (Kipf & Welling, 2017; Hamilton et al., 2017; Veličković et al., 2018; Xu et al., 2019), most of them falling under a general Message Passing Neural Networks (MPNNs) framework (Gilmer et al., 2017). MPNNs learn node representations by an iterative neighborhood-aggregation process, where each layer updates each node's representation by combining previous-layer representations of the node itself and its neighbors. The node feature vector is used as the initial node representation. Thus, MPNNs combine node features with graph topology, allowing them to learn complex dependencies between nodes.

In many real-world networks, edges tend to connect similar nodes. This property is called *homophily*. Typical examples of homophilous networks are social networks, where users tend to connect to users with similar interests, and citation networks, where papers mostly cite works from the

same research area. The opposite of homophily is called *heterophily*: this property describes the preference of network nodes to connect to nodes not similar to them. For example, in financial transaction networks, fraudsters often perform transactions with non-fraudulent users, and in dating networks, most connections are between people of opposite genders.

Early works on GNNs mostly evaluated their models on homophilous graphs. This has led to claims that GNNs implicitly use the homophily of a graph and are thus not suitable for heterophilous datasets (Zhu et al., 2021; 2020; He et al., 2022; Wang et al., 2022). Recently, many works have proposed new GNN models specifically designed for heterophilous graphs that are claimed to outperform standard GNNs. However, these models are typically evaluated on the same six heterophilous graphs first used in the context of learning under heterophily by Pei et al. (2020). In this work, we challenge this evaluation setting. We highlight several downsides of the standard heterophilous datasets, such as low diversity, small size, extreme class imbalance of some datasets, and, most importantly, the presence of a large number of duplicate nodes in `squirrel` and `chameleon` datasets. We show that models rely on the train-test data leakage introduced by duplicated nodes to achieve strong results, and removing these nodes significantly affects the performance of the models.

Motivated by the shortcomings of the currently used heterophilous benchmarks, we collect a set of diverse heterophilous graphs and propose to use them as a better benchmark. The proposed datasets come from different domains and exhibit a variety of structural properties. We evaluate a wide range of GNNs, both standard and heterophily-specific, on the proposed benchmark, which, to the best of our knowledge, constitutes the most extensive empirical study of heterophily-specific models. In doing so, we uncover that the standard baselines almost always outperform heterophily-specific models. Thus, the progress in learning under heterophily might have been limited to the standard datasets used for evaluation. Our results also show that there is, however, a trick that is useful for learning on heterophilous graphs — separating ego- and neighbor-embeddings, which was proposed in Zhu et al. (2020). This trick consistently improves the baselines (such as GAT and Graph Transformer) and allows one to achieve the best results. We hope that the proposed benchmark will be helpful for further progress in learning under heterophily.

## 2 RELATED WORK

**Measuring homophily**   While much effort has been put into developing graph representation learning methods for heterophilous graphs, there is no universally agreed-upon measure of homophily. Homophily measures typically used in the literature are *edge homophily* (Abu-El-Haija et al., 2019; Zhu et al., 2020), which is simply the fraction of edges that connect nodes of the same class, and *node homophily* (Pei et al., 2020), which computes the proportion of neighbors that have the same class for each node and then averages these values across all nodes. These two measures are simple and intuitive; however, as shown in Lim et al. (2021); Platonov et al. (2022), they are sensitive to the number of classes and their balance, which makes these measures hard to interpret and incomparable across different datasets. To fix these issues, Lim et al. (2021) propose another homophily measure. However, Platonov et al. (2022) show that it also can provide unreliable results. To solve the issues with existing measures, Platonov et al. (2022) propose to use *adjusted homophily*, which corrects the number of intra-class edges by their expected value. Thus, adjusted homophily becomes insensitive to the number of classes and their balance. Platonov et al. (2022) show that adjusted homophily satisfies a number of desirable properties, which makes it appropriate for comparing homophily levels between different datasets. Thus, in our work, we will use adjusted homophily for measuring homophily of graphs.

**Graph datasets**   Early works on GNNs mostly evaluated their models on highly homophilous graphs. The most popular of them are three citation networks: `cora`, `citeseer`, and `pubmed` (Giles et al., 1998; McCallum et al., 2000; Namata et al., 2012; Sen et al., 2008; Yang et al., 2016). Examples of other graph datasets for node classification that appear in the literature include citation networks `coauthor-cs`, `coauthor-physics` and co-purchasing networks `amazon-computers`, `amazon-photo` from Shchur et al. (2018), discussion network `reddit` from Hamilton et al. (2017). These datasets also have high levels of homophily. Recently, Open Graph Benchmark (Hu et al., 2020) was created to provide challenging large-scale graphs for evaluating GNN performance. The proposed datasets such as `ogbn-arxiv`, `ogbn-products`, `ogbn-papers100M` are also highly homophilous (Zhu et al., 2020; Platonov et al., 2022).

As for heterophilous graphs, the datasets used in most studies dedicated to learning under heterophily are limited to the six graphs adopted by Pei et al. (2020): `squirrel`, `chameleon`, `actor`, `texas`, `cornell`, and `wisconsin`. These graphs have become the de-facto standard benchmark for evaluating heterophily-specific models and were used in numerous papers (Zhu et al., 2021; 2020; Chien et al., 2021; Yan et al., 2022; Maurya et al., 2022; Li et al., 2022; He et al., 2022; Wang & Zhang, 2022; Wang et al., 2022; Du et al., 2022; Suresh et al., 2021; Bo et al., 2021; Ma et al., 2022; Luan et al., 2022; Bodnar et al., 2022). We further discuss these datasets in Section 3. Recently, a set of large-scale heterophilous graph datasets has been proposed in Lim et al. (2021). However, due to their size, these datasets are primarily suitable for evaluating scalable graph methods rather than GNNs, and thus have not seen a wide adoption in the GNN community yet.

**Specialized methods for learning under heterophily** Many methods designed for achieving good results either specifically under heterophily or in both homophily and heterophily settings have been recently proposed. In this paragraph, we briefly describe some of them. Pei et al. (2020) were the first to attract attention to learning under heterophily. Their approach (Geom-GCN) precomputes unsupervised node embeddings and defines convolution in the latent space of these embeddings. Zhu et al. (2020) is another pioneering work on heterophily that identifies three designs in existing GNNs that allow the models to generalize to the heterophily setting: ego- and neighbor-embedding separation, aggregation across higher-order neighborhoods, and combining intermediate representations from different layers for the final node representation. Zhu et al. (2021) further proposed a new architecture CPGNN that incorporates a learnable class compatibility matrix in the GNN aggregation step that can model different levels of homophily. Chien et al. (2021) developed a Generalized PageRank-inspired architecture (GPR-GNN) with learnable weights designed to adapt to various node label patterns. Yan et al. (2022) related learning under heterophily to the problem of oversmoothing and suggested two modifications to the GCN architecture: degree corrections and signed messages. Maurya et al. (2022) proposed FSGNN which decouples feature aggregation from GNN layers and uses soft feature selection. Li et al. (2022) developed GloGNN and GloGNN++ models that aggregate information from global nodes in the graph. He et al. (2022) proposed block-modeling guided GNN architecture that can learn different aggregation rules for different nodes. Wang & Zhang (2022) introduced JacobiConv — a spectral GNN that is supposed to achieve strong results on both homophilous and heterophilous graphs. Wang et al. (2022) designed a new propagation mechanism that can adaptively change propagation and aggregation for different nodes. Du et al. (2022) proposed GBK-GNN that uses bi-kernel feature transformation and a selection gate to capture useful information in both homophily and heterophily settings. Suresh et al. (2021) suggested transforming the input graph into a computation graph based on both proximity and structural information. Bo et al. (2021) proposed a self-gating mechanism that allows their model to adaptively integrate both low-frequency and high-frequency signals. Luan et al. (2022) introduced the Adaptive Channel Mixing (ACM) framework to address those cases of heterophily that are harmful for GNN performance. Bodnar et al. (2022) proposed neural sheaf diffusion models that learn cellular sheaves from data to achieve strong results on heterophilous graphs.

**Performance of standard GNNs under heterophily** While it is widely considered that standard GNNs do not perform well under heterophily (Zhu et al., 2021; 2020; He et al., 2022; Wang et al., 2022), recently, there have been several works that show that standard GNNs can achieve strong results on some heterophilous graphs (Ma et al., 2022; Luan et al., 2022). However, these results were primarily obtained on synthetic or semi-synthetic datasets. Platonov et al. (2022) explain these observations by high *label informativeness* of the considered graphs: mutual information between neighbors' labels can be high even when these neighbors have different labels. We show that standard GNNs also often achieve strong results and outperform specialized methods on real-world graphs with low label informativeness.

## 3 ISSUES WITH POPULAR HETEROPHILOUS DATASETS

In this section, we revisit datasets commonly used for heterophilous node classification. As discussed in Section 2, the following six datasets are the most popular: Wikipedia networks `squirrel` and `chameleon`, actor co-occurrence in Wikipedia pages network (`actor`), and WebKB datasets `texas`, `wisconsin`, and `cornell`. The standard preprocessing of these datasets is done by Pei et al. (2020). First, we note that these datasets only come from three sources; thus, they do not

provide sufficient coverage of different heterophilous patterns that can be found in real data, and more diverse datasets are required for a fair evaluation of models under heterophily. However, that is not the only problem with this benchmark. In this section, we show that some of these datasets have certain drawbacks that may highly affect the evaluation results.

## 3.1 SQUIRREL AND CHAMELEON

These datasets are initially collected by Rozemberczki et al. (2021): nodes represent articles from the English Wikipedia (December 2018), and edges reflect mutual links between them. Node features indicate the presence of particular nouns in the articles. The target variable is the average monthly traffic for the web page, and the task is node regression. Pei et al. (2020) converted the task to node classification by grouping nodes into five categories based on the original regression target, and this preprocessing became standard in the literature.

While analyzing these datasets, we noticed many groups of nodes with exactly the same regression target and exactly the same neighborhood. For instance, in `squirrel` there is a group of 48 nodes that all have the same regression target 370193 and the same 15 neighbors, and in `chameleon` there is a group of 92 nodes with the same regression target 14480 and the same 18 neighbors. For brevity, we further call such nodes 'duplicates'. We note that while it is expected for some nodes in a natural graph to have the same neighborhood, it is highly unlikely for many nodes to share the same average monthly traffic (which in these datasets is an integer in the range 0-850K). However, not only do some nodes in these datasets simultaneously share the same regression target and neighborhood, but the number of such duplicates is very large. Since duplicates from the same group appear in the train, validation, and test parts of the datasets, they create a train-test data leakage: for duplicates from the test set, their labels can be predicted by simply matching the node's neighborhood to neighborhoods of train nodes. This leakage is present not only for the original node regression task, but also for the node classification task, since labels for the classification task are based on the regression target. We further show that removing this data leakage strongly affects the performance of GNNs.

Upon further investigation of these datasets, we found the following: 1) the duplicate nodes may have different features, 2) all the edges of the duplicates are outgoing, 3) for (almost) every such group of duplicates, there is a unique node in the dataset with the same average monthly traffic and the same outgoing edges, but with some additional incoming edges. We hypothesize that this can be the actual version of the web page that should be present in the dataset, while all the other nodes with the same average monthly traffic and the same outgoing edges should be removed.

Table 1 shows the number of nodes in `squirrel` and `chameleon`, as well as the number of duplicates and non-duplicates. The duplicates constitute more than half of each dataset. In the same table, we report the accuracy of Graph-SAGE (Hamilton et al., 2017) on duplicates and non-duplicates separately. We can see a signif-

Table 1: Duplicates in Wikipedia datasets

|  | squirrel | chameleon |
|---|---|---|
| number of nodes | 5201 | 2277 |
| number of duplicates | 2978 | 1387 |
| number of non-duplicates | 2223 | 890 |
| GraphSAGE accuracy | | |
| on duplicates | $51.69 \pm 01.68$ | $74.89 \pm 02.05$ |
| on non-duplicates | $34.67 \pm 02.30$ | $46.17 \pm 03.21$ |

icant difference in performance on these two types of nodes, confirming that the model implicitly relies on data leakage to make predictions. We additionally note that duplicates are present in all classes and provide the distribution of duplicates across classes in Table 5 in Appendix B.

We further filtered these datasets by removing all duplicates, i.e., nodes with no incoming edges and for which there exists a node in the graph with the same average monthly traffic and the same set of outgoing edges. We evaluated several models on the original and filtered datasets; see Table 2 for the results (we refer to Section 5.1 and Appendix A for the description of the models and the evaluation setup). First, we see a significant performance drop for many models, particularly on the `chameleon` dataset. This performance drop confirms that the models implicitly rely on the leaked data to achieve strong results on unfiltered datasets. Moreover, we see that the exact drop in performance significantly differs between models, and thus the ranking of the models on the filtered datasets is very different from the ranking on the original datasets. This suggests that different

Table 2: Accuracy of models on original and filtered `squirrel` and `chameleon`. The 'ranks' columns report the positions in the ranked list of models on the original and filtered datasets.

| | squirrel accuracy on original dataset | accuracy on filtered dataset | ranks | chameleon accuracy on original dataset | accuracy on filtered dataset | ranks |
|---|---|---|---|---|---|---|
| ResNet | $33.88 \pm 1.79$ | $36.55 \pm 1.82$ | 12 / 7 | $49.52 \pm 1.73$ | $36.73 \pm 4.71$ | 12 / 14 |
| ResNet+SGC | $34.36 \pm 1.21$ | $38.36 \pm 1.97$ | 11 / 5 | $49.93 \pm 2.27$ | $41.01 \pm 4.54$ | 11 / 2 |
| ResNet+adj | $65.46 \pm 1.58$ | $38.37 \pm 1.99$ | 2 / 4 | $71.07 \pm 2.23$ | $38.67 \pm 3.87$ | 2 / 12 |
| GCN | $39.06 \pm 1.52$ | $39.47 \pm 1.47$ | 6 / 2 | $50.18 \pm 3.29$ | $40.89 \pm 4.12$ | 10 / 3 |
| SAGE | $35.83 \pm 1.32$ | $36.09 \pm 1.99$ | 9 / 9 | $50.18 \pm 1.78$ | $37.77 \pm 4.14$ | 9 / 13 |
| GAT | $32.21 \pm 1.63$ | $35.62 \pm 2.06$ | 14 / 11 | $45.02 \pm 1.75$ | $39.21 \pm 3.08$ | 16 / 9 |
| GAT-sep | $35.72 \pm 1.98$ | $35.46 \pm 3.10$ | 10 / 13 | $50.24 \pm 2.22$ | $39.26 \pm 2.50$ | 8 / 8 |
| GT | $31.61 \pm 1.10$ | $36.30 \pm 1.98$ | 15 / 8 | $44.93 \pm 1.40$ | $38.87 \pm 3.66$ | 17 / 11 |
| GT-sep | $36.08 \pm 1.58$ | $36.66 \pm 1.63$ | 8 / 6 | $50.33 \pm 2.57$ | $40.31 \pm 3.01$ | 7 / 5 |
| H$_2$GCN | $29.45 \pm 1.65$ | $35.10 \pm 1.15$ | 17 / 15 | $46.27 \pm 2.71$ | $26.75 \pm 3.64$ | 15 / 16 |
| CPGNN | $30.91 \pm 1.98$ | $30.04 \pm 2.03$ | 16 / 16 | $48.77 \pm 2.10$ | $33.00 \pm 3.15$ | 13 / 15 |
| GPR-GNN | $33.39 \pm 2.05$ | $38.95 \pm 1.99$ | 13 / 3 | $47.26 \pm 1.74$ | $39.93 \pm 3.30$ | 14 / 6 |
| FSGNN | $68.93 \pm 1.69$ | $35.92 \pm 1.32$ | 1 / 10 | $77.85 \pm 0.46$ | $40.61 \pm 2.97$ | 1 / 4 |
| GloGNN | $61.21 \pm 1.96$ | $35.11 \pm 1.24$ | 3 / 14 | $70.04 \pm 2.12$ | $25.90 \pm 3.58$ | 3 / 17 |
| FAGCN | $47.63 \pm 1.85$ | $41.08 \pm 2.27$ | 4 / 1 | $64.23 \pm 2.04$ | $41.90 \pm 2.72$ | 5 / 1 |
| GBK-GNN | $37.06 \pm 1.24$ | $35.51 \pm 1.65$ | 7 / 12 | $51.36 \pm 1.79$ | $39.61 \pm 2.60$ | 6 / 7 |
| JacobiConv | $46.17 \pm 4.34$ | $29.71 \pm 1.66$ | 5 / 17 | $68.33 \pm 1.38$ | $39.00 \pm 4.20$ | 4 / 10 |

models have different capacity to utilize the data leakage. To better illustrate the difference in rankings, we report model ranks on both original and filtered datasets in Table 2. Some models have particularly strong performance changes. For example, FSGNN is the best model on both original datasets, however, on filtered `squirrel` and `chameleon` it achieves only 10th and 4th places, respectively. Such a substantial shake-up raises concerns about the validity of conclusions made in previous works that rely on analyzing the performance of different models on these datasets.

## 3.2 CORNELL, TEXAS, WISCONSIN

`Cornell`, `texas`, and `wisconsin` were introduced by Pei et al. (2020). These are three sub-datasets of the WebKB[1] webpage dataset collected from computer science departments of various universities. In these datasets, nodes are web pages, and edges are hyperlinks between them. Node features are the bag-of-words representation of the web pages. The target is the web page category: 'student', 'project', 'course', 'staff', or 'faculty'. We first note that these datasets are very small (183-251 nodes and 295-499 edges), which can lead to unstable and statistically insignificant results. Indeed, from the results of various models reported in previous works, it can be seen that the standard deviation on these datasets is very high. Moreover, these datasets have very imbalanced classes, to the point that the `texas` dataset has a class that consists of only one node, which makes using this class for training and evaluation meaningless. We report the number of nodes in different classes of these datasets in Table 6 in Appendix B. We note that all the previous works that use these datasets report accuracy on them, however, this metric is not designed for measuring performance under strong class imbalance and can provide misleading results in this setting.

## 4 NEW HETEROPHILOUS DATASETS

Motivated by the observations described in the previous section, we collected several new datasets for evaluating GNNs under heterophily. We aim to obtain a set of datasets satisfying the following conditions:

- Datasets should be heterophilous. We evaluate this using the adjusted homophily measure; see the formal definition below.

---

[1]http://www.cs.cmu.edu/afs/cs.cmu.edu/project/theo-11/www/wwkb

- Graph structure should be helpful for the task. To verify this, we compare the performance of graph-agnostic ResNet with GNN methods. We expect GNNs to have a noticeable gain in performance.

- Datasets should be diverse, i.e., come from various domains and have different structural properties. Thus, for each dataset, we report several characteristics that we describe below.

- The size of the graphs should be large enough to provide statistically significant results, but small enough to allow for evaluating most of the heterophily-specific models proposed in the literature, which are often non-scalable. Thus, we only collect graphs that have 10K-50K nodes.

For each of the proposed datasets, we report its basic characteristics, such as the number of nodes, edges, features, and classes, as well as various graph statistics, which we now define.

First, we measure homophily. As discussed above, we focus on adjusted homophily, but we also report edge homophily to be comparable with previous studies reporting this measure. However, we emphasize that edge homophily does not produce meaningful results for datasets with unbalanced classes which are present in our benchmark. Formally, edge homophily is

$$h_{edge} = \frac{|(u,v) \in E : y_u = y_v\}|}{|E|},$$

where $y_u$ is the label of a node $u$ and $E$ is the set of edges. Adjusted homophily is based on the edge homophily and can be computed as follows:

$$h_{adj} = \frac{h_{edge} - \sum_{k=1}^{C} D_k^2/(2|E|)^2}{1 - \sum_{k=1}^{C} D_k^2/(2|E|)^2},$$

where $D_k := \sum_{v : y_v = k} d(v)$ and $d(v)$ denotes the degree of a node $v$. In Platonov et al. (2022) it was shown that adjusted homophily satisfies a number of desirable properties, which makes it appropriate for comparing datasets with different number of classes and class size balance.

We also report *label informativeness* (LI) introduced in Platonov et al. (2022) and shown to better agree with GNN performance than homophily. Label informativeness quantifies how much information a neighbor's label gives about the node's label. To formally define this measure, we let $(\xi, \eta) \in E$ be an edge sampled uniformly at random among all edges and define

$$\text{LI} := I(y_\xi, y_\eta)/H(y_\xi).$$

Here $y_\xi$ and $y_\eta$ are (random) labels of $\xi$ and $\eta$, $H(y_\xi)$ is the entropy of $y_\xi$ and $I(y_\xi, y_\eta)$ is the mutual information of $\xi$ and $\eta$.

We also report several standard graph characteristics, such as diameter and clustering coefficient. In the literature, there are two popular definitions of the clustering coefficient (Boccaletti et al., 2014). The global clustering coefficient is the ratio between the number of triangles and the number of pairs of adjusted edges. To get the average local clustering coefficient, we first compute the clustering for each node and then average the obtained values across all nodes.

Table 3 provides statistics of the five new datasets we propose for evaluating GNN performance under heterophily. One can see that these datasets have diverse properties. `Tolokers` is the densest graph with an average node degree above 88, while the rest of the graphs are sparse, `roman-empire` being the sparsest one. `Questions` has very low values of clustering coefficients compared to other graphs, which shows that it has a small proportion of closed node triplets. `Roman-empire` is the only graph in our benchmark with a value of label informativeness significantly larger than zero. Below we will describe each of the new datasets in more detail.

**Roman-empire** This dataset is based on the Roman Empire article from English Wikipedia, which was selected since it is one of the longest articles on Wikipedia. The text was retrieved from the English Wikipedia 2022.03.01 dump from Lhoest et al. (2021). Each node in the graph corresponds to one (non-unique) word in the text. Thus, the number of nodes in the graph is equal to the article's length. Two words are connected with an edge if at least one of the following two conditions holds: either these words follow each other in the text, or these words are connected in the dependency tree of the sentence (one word depends on the other). Thus, the graph is a chain graph with additional

Table 3: Statistics of the new heterophilous datasets

|  | roman-empire | amazon-ratings | minesweeper | tolokers | questions |
|---|---|---|---|---|---|
| nodes | 22662 | 24492 | 10000 | 11758 | 48921 |
| edges | 32927 | 93050 | 39402 | 519000 | 153540 |
| avg degree | 2.91 | 7.60 | 7.88 | 88.28 | 6.28 |
| global clustering | 0.29 | 0.32 | 0.43 | 0.23 | 0.02 |
| avg local clustering | 0.39 | 0.58 | 0.44 | 0.53 | 0.03 |
| diameter | 6824 | 46 | 99 | 11 | 16 |
| node features | 300 | 300 | 7 | 10 | 301 |
| classes | 18 | 5 | 2 | 2 | 2 |
| edge homophily | 0.05 | 0.38 | 0.68 | 0.59 | 0.84 |
| adjusted homophily | -0.05 | 0.14 | 0.01 | 0.09 | 0.02 |
| LI | 0.11 | 0.04 | 0.00 | 0.01 | 0.00 |

shortcut edges corresponding to syntactic dependencies between words. The class of a node is its syntactic role (we select the 17 most frequent roles as unique classes and group all the other roles into the 18th class). The syntactic roles were obtained using spaCy (Honnibal et al., 2020). For node features, we use FastText word embeddings (Grave et al., 2018). While this task can probably be better solved with models from the field of NLP, we adapt it to evaluate GNNs in the setting of low homophily, sparse connectivity, and potential long-range dependencies.

This graph has 22.7K nodes and 32.9K edges. By construction, the structure of this graph is chain-like; thus, it has the smallest average degree (2.9) and the largest diameter (6824). This graph is heterophilous, $h_{adj} = -0.05$. Interestingly, this dataset has a larger value of label informativeness compared to all the other heterophilous datasets analyzed by Platonov et al. (2022). This means that there are non-trivial label connectivity patterns specific to this dataset.

**Amazon-ratings** This dataset is based on the Amazon product co-purchasing network metadata dataset[2] from SNAP Datasets (Leskovec & Krevl, 2014). Nodes are products (books, music CDs, DVDs, VHS video tapes), and edges connect products that are frequently bought together. The task is to predict the average rating given to a product by reviewers. We grouped possible rating values into five classes. For node features, we use the mean of FastText embeddings (Grave et al., 2018) for words in the product description. To reduce the size of the graph, we only consider the largest connected component of the 5-core of the graph.

**Minesweeper** This dataset is inspired by the Minesweeper game, and it is the only synthetic dataset in our benchmark. The graph is a regular 100x100 grid where each node (cell) is connected to eight neighboring nodes (with the exception of nodes at the edge of the grid, which have fewer neighbors). 20% of the nodes are randomly selected as mines. The task is to predict which nodes are mines. The node features are one-hot-encoded numbers of neighboring mines. However, for randomly selected 50% of the nodes, the features are unknown, which is indicated by a separate binary feature.

The structure of this graph is significantly different from the other datasets due to its regularity. The average degree is 7.88 since almost all the nodes have exactly eight neighbors. Since mines are placed randomly, both adjusted homophily and label informativeness are close to zero.

**Tolokers** This dataset is based on data from the Toloka crowdsourcing platform (Likhobaba et al., 2023). The nodes represent tolokers (workers) that have participated in at least one of 13 selected projects. An edge connects two tolokers if they have worked on the same task. The goal is to predict which tolokers have been banned in one of the projects. Node features are based on the worker's profile information and task performance statistics.

This graph has 11.8K nodes, with the average degree of 88.28. Thus, the graph is significantly denser than all the other graphs. About 22% of the tolokers in this dataset have been banned.

---

[2]https://snap.stanford.edu/data/amazon-meta.html

**Questions** This dataset is based on data from the question-answering website Yandex Q. Nodes are users, and an edge connects two nodes if one user answered the other user's question during a one-year time interval (from September 2021 to August 2022). To restrict the size of the dataset, we consider only users interested in the topic 'medicine'. The task is to predict which users remained active on the website (were not deleted or blocked) at the end of the period. For node features, we use the mean of FastText embeddings (Grave et al., 2018) for words in the user description. Since some users (15%) do not have descriptions, we use an additional binary feature that indicates such users.

The obtained dataset has 48.9K nodes, and the average degree is 6.28. We note that the classification task is highly unbalanced: 97% of the users are in the active class. This causes high edge homophily, but the adjusted homophily indicates that the graph is heterophilous: $h_{adj} = 0.02$. This dataset has the smallest clustering coefficients among the proposed ones, which means it has a small fraction of closed node triplets.

## 5 BENCHMARKING EXISTING ALGORITHMS

### 5.1 SETUP

**Baselines** We choose several representative neural architectures as our baselines. First, we use a **ResNet**-like model (He et al., 2016) as a graph-agnostic baseline. This model treats all nodes as independent samples and does not have access to the graph topology. Thus, if graph topology provides useful information for the task, we expect other models to outperform ResNet. Further, we use two simple node feature augmentation strategies to provide ResNet with some information about the graph structure. One strategy is multiplying the initial node feature matrix with a power of normalized graph adjacency matrix, which smooths node features along graph edges. This approach was proposed in Wu et al. (2019) (their proposed model SGC is a linear classifier on top of the preprocessed features, while we use a ResNet-like model instead of the linear classifier). We name this model **ResNet+SGC**. Another strategy is augmenting node features with the rows of the adjacency matrix, thus directly providing information about the graph connectivity. This approach is inspired by LINK (Zheleva & Getoor, 2009) — a linear model using the adjacency matrix rows as features — and is very similar to the recently proposed LINKX model (Lim et al., 2021), which also combines node features and adjacency matrix rows, but uses a custom model. We name this version of the model **ResNet+adj**.

Further, we use 2 classic GNN architectures: **GCN** (Kipf & Welling, 2017) and **Graph-SAGE** (Hamilton et al., 2017). For GraphSAGE, we use the version with the mean aggregation function and do not use the node sampling technique used in the original paper.

As a more advanced GNN architecture, we take **GAT** (Veličković et al., 2018), which uses attention-based aggregation. However, GAT uses a very simple attention mechanism and, as a result, can only compute a limited kind of attention — for instance, the ranking of the attention scores does not depend on the query node (Brody et al., 2022). To overcome this limitation, we also use a model with a more powerful attention mechanism — Graph Transformer (**GT**) (Shi et al., 2020), which is an adaptation of the popular Transformer architecture (Vaswani et al., 2017) to graphs. Note that in this version of GT, each node can only attend to its neighbors.

Zhu et al. (2020) shows that separating ego- and neighbor-embeddings in the GNN aggregation step (as done in GraphSAGE, where the node's embedding is concatenated to the mean of its neighbors' embeddings instead of being summed with them) is beneficial when learning under heterophily. Thus, we add this simple architectural modification to GAT and GT models, which originally do not separate ego- and neighbor embeddings. We name these model modifications **GAT-sep** and **GT-sep**.

We augment all our baseline models with skip connections (He et al., 2016) and layer normalization (Ba et al., 2016), which are standard neural architecture elements in modern deep learning. We found these techniques to be important for the strong performance of our baselines.

**Heterophily-specific models** We use eight models designed for node classification under heterophily: **H₂GCN** (Zhu et al., 2020), **CPGNN** (Zhu et al., 2021), **GPR-GNN** (Chien et al., 2021), **FSGNN** (Maurya et al., 2022), **GloGNN** (Li et al., 2022), **FAGCN** (Bo et al., 2021), **GBK-GNN** (Du

Table 4: The performance of models on the proposed datasets. Accuracy is reported for `roman-empire` and `amazon-ratings`, and ROC AUC is reported for `minesweeper`, `tolokers`, and `questions`.

|  | roman-empire | amazon-ratings | minesweeper | tolokers | questions |
|---|---|---|---|---|---|
| ResNet | $65.88 \pm 0.38$ | $45.90 \pm 0.52$ | $50.89 \pm 1.39$ | $72.95 \pm 1.06$ | $70.34 \pm 0.76$ |
| ResNet+SGC | $73.90 \pm 0.51$ | $50.66 \pm 0.48$ | $70.88 \pm 0.90$ | $80.70 \pm 0.97$ | $75.81 \pm 0.96$ |
| ResNet+adj | $52.25 \pm 0.40$ | $51.83 \pm 0.57$ | $50.42 \pm 0.83$ | $78.78 \pm 1.11$ | $75.77 \pm 1.24$ |
| GCN | $73.69 \pm 0.74$ | $48.70 \pm 0.63$ | $89.75 \pm 0.52$ | $83.64 \pm 0.67$ | $76.09 \pm 1.27$ |
| SAGE | $85.74 \pm 0.67$ | $53.63 \pm 0.39$ | $93.51 \pm 0.57$ | $82.43 \pm 0.44$ | $76.44 \pm 0.62$ |
| GAT | $80.87 \pm 0.30$ | $49.09 \pm 0.63$ | $92.01 \pm 0.68$ | $83.70 \pm 0.47$ | $77.43 \pm 1.20$ |
| GAT-sep | $88.75 \pm 0.41$ | $52.70 \pm 0.62$ | $93.91 \pm 0.35$ | $83.78 \pm 0.43$ | $76.79 \pm 0.71$ |
| GT | $86.51 \pm 0.73$ | $51.17 \pm 0.66$ | $91.85 \pm 0.76$ | $83.23 \pm 0.64$ | $77.95 \pm 0.68$ |
| GT-sep | $87.32 \pm 0.39$ | $52.18 \pm 0.80$ | $92.29 \pm 0.47$ | $82.52 \pm 0.92$ | $78.05 \pm 0.93$ |
| $H_2$GCN | $60.11 \pm 0.52$ | $36.47 \pm 0.23$ | $89.71 \pm 0.31$ | $73.35 \pm 1.01$ | $63.59 \pm 1.46$ |
| CPGNN | $63.96 \pm 0.62$ | $39.79 \pm 0.77$ | $52.03 \pm 5.46$ | $73.36 \pm 1.01$ | $65.96 \pm 1.95$ |
| GPR-GNN | $64.85 \pm 0.27$ | $44.88 \pm 0.34$ | $86.24 \pm 0.61$ | $72.94 \pm 0.97$ | $55.48 \pm 0.91$ |
| FSGNN | $79.92 \pm 0.56$ | $52.74 \pm 0.83$ | $90.08 \pm 0.70$ | $82.76 \pm 0.61$ | $78.86 \pm 0.92$ |
| GloGNN | $59.63 \pm 0.69$ | $36.89 \pm 0.14$ | $51.08 \pm 1.23$ | $73.39 \pm 1.17$ | $65.74 \pm 1.19$ |
| FAGCN | $65.22 \pm 0.56$ | $44.12 \pm 0.30$ | $88.17 \pm 0.73$ | $77.75 \pm 1.05$ | $77.24 \pm 1.26$ |
| GBK-GNN | $74.57 \pm 0.47$ | $45.98 \pm 0.71$ | $90.85 \pm 0.58$ | $81.01 \pm 0.67$ | $74.47 \pm 0.86$ |
| JacobiConv | $71.14 \pm 0.42$ | $43.55 \pm 0.48$ | $89.66 \pm 0.40$ | $68.66 \pm 0.65$ | $73.88 \pm 1.16$ |

et al., 2022), and **JacobiConv** (Wang & Zhang, 2022). To the best of our knowledge, this is the most extensive comparison of heterophily-specific models in the literature.

We provide details about our training setup and hyperparameter selection in Appendix A.

## 5.2 RESULTS

Table 4 shows the performance of different models on our datasets. We can see that the best results are almost always achieved by baselines rather than heterophily-specific models. Among 15 of the top-3 performances on our 5 datasets, 13 belong to standard GNNs. Occasionally, some heterophily-specific models perform even worse than the graph-agnostic ResNet baseline. These results show that the progress in learning under heterophily made in recent years was limited to the particular datasets used for evaluation, while we observe that standard GNNs generally outperform specialized models. The only specialized model that consistently achieves strong performance and occasionally reaches top-3 best results is FSGNN, a simple model often overlooked in the literature.

As for standard GNNs, we notice that the best results are almost always achieved by models that separate ego- and neighbor-embeddings (GraphSAGE, GAT-sep, GT-sep). GAT-sep and GT-sep typically outperform their versions without embedding separation, which shows that this trick proposed in Zhu et al. (2020) is indeed helpful for learning under heterophily.

## 6 CONCLUSION

In this paper, we uncover significant problems with the datasets typically used to evaluate the performance of GNNs under heterophily. The most sifnificant of these problems is the presence of a large number of duplicate nodes in `squirrel` and `chameleon` datasets, which leads to a train-test data leakage. We show that the removal of these duplicates drastically changes the relative performance of different models.

Motivated by this issue, we propose several new heterophilous datasets of different nature and with diverse structural properties that can form a better benchmark. We evaluate a variety of standard GNNs and heterophily-specific models on these datasets and show that standard GNNs generally outperform specialized models. We hope that the proposed benchmark will be useful for further progress in learning under heterophily.

ACKNOWLEDGMENTS

We thank Daniil Likhobaba, Nikita Pavlichenko, and Dmitry Ustalov for providing the `tolokers` dataset. We also thank Alexandr Andreev and Irina Lialikova for collecting the Yandex Q data for the `questions` dataset.

The publication was partly supported by the grant for research centers in the field of AI provided by the Analytical Center for the Government of the Russian Federation (ACRF) in accordance with the agreement on the provision of subsidies (identifier of the agreement 000000D730321P5Q0002) and the agreement with HSE University No. 70-2021-00139.

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

Table 5: Distribution of duplicates across classes in Wikipedia datasets

|  | class 1 | class 2 | class 3 | class 4 | class 5 |
|---|---|---|---|---|---|
|  |  |  | squirrel |  |  |
| number of nodes | 1042 | 1040 | 1039 | 1040 | 1040 |
| number of duplicates | 286 | 524 | 642 | 719 | 807 |
| number of non-duplicates | 756 | 516 | 397 | 321 | 233 |
|  |  |  | chameleon |  |  |
| number of nodes | 456 | 460 | 453 | 521 | 387 |
| number of duplicates | 214 | 326 | 244 | 357 | 246 |
| number of non-duplicates | 242 | 134 | 209 | 164 | 141 |

## A  TRAINING DETAILS AND HYPERPARAMETERS SELECTION

In this section, we describe the details of our training setup for experiments in Section 3.1 and Section 5. For `squirrel` and `chameleon`, we use the 10 existing standard train/validation/test splits. For filtered versions of these datasets, we use the same splits with duplicates removed. For each of our new proposed datasets, we fix 10 random 50%/25%/25% train/validation/test splits. We train each model on each split once, reporting mean performance and standard deviation. For multiclass classification datasets (`roman-empire`, `amazon-ratings`) we report accuracy, and for binary classification datasets (`minesweeper`, `tolokers`, `questions`) we report ROC AUC.

The `squirrel` and `chameleon` datasets are directed. Most codebases implementing heterophily-specific models do not convert these graphs to undirected; therefore, we also treat them as directed. In contrast, all the graphs in our proposed benchmark are undirected.

We found that our baseline models are quite robust to hyperparameter values, so the only hyperparameter that we tune for them is the number of layers. We choose it from the set $\{1, 2, 3, 4, 5\}$ based on the validation performance. For all the other hyperparameters, we use the same values across all baseline models and datasets. Namely, we use the following hyperparameter values: the hidden dimension is 512, and the dropout probability is 0.2. For GAT and Graph Transformer models, the number of attention heads is set to 8. We use GELU activation functions (Hendrycks & Gimpel, 2016) in all our baseline models. We use the Adam optimizer (Kingma & Ba, 2015) with learning rate of $3 \cdot 10^{-5}$. We train each model for 1000 steps and select the best step based on the performance on the validation set. Our baselines are implemented using PyTorch (Paszke et al., 2019) and DGL (Wang et al., 2019).

For heterophily-specific models, we use the official code provided by the authors of these models. Unlike the baselines, heterophily-specific models turned out to be quite sensitive to the particular choice of hyperparameters. Namely, the choice of learning rate and weight decay may significantly impact the model performance. For different models, the range of optimal hyperparameter values may differ drastically. Therefore, for each model, we have searched over a specific hyperparameter grid. Many models under consideration have their specific hyperparameters. We have fixed them to the values set for the `squirrel` dataset in all cases except for the GloGNN model (Li et al., 2022), which turned out to be very sensitive to its specific hyperparameters. Models are trained for the same number of steps as in the original papers, and we use early stopping on the validation set with the patience of 100 steps to prevent overfitting. For each model, we swept over 4-5 values of learning rate and weight decay and selected the one with the best validation performance.

## B  ADDITIONAL DATASET STATISTICS

In Table 5, we show the distribution of duplicates across classes in the `squirrel` and `chameleon` datasets. We can see that there is a large number of duplicates in all classes, however, their distribution is not even.

In Table 6, we report the distribution of nodes across classes in the `texas`, `cornell`, and `wisconsin` datasets.

Table 6: Number of nodes in different classes of `texas`, `cornell`, and `wisconsin`

|  | class 1 | class 2 | class 3 | class 4 | class 5 |
|---|---|---|---|---|---|
| texas | 33 | 1 | 18 | 101 | 30 |
| cornell | 38 | 16 | 30 | 82 | 17 |
| wisconsin | 10 | 70 | 118 | 32 | 21 |

## C  COMPARISON TO THE BENCHMARK PROPOSED IN LIM ET AL. (2021)

Recently, a benchmark of large-scale heterophilous graph datasets has been proposed by Lim et al. (2021). This section describes how this benchmark differs from our proposed datasets. The first difference is the size of the graphs. Lim et al. (2021) specifically collect large datasets to evaluate the performance of scalable graph methods under heterophily. However, this prevents them from comparing to many GNNs designed for heterophilous graphs since such GNNs are often compute and memory intensive and thus cannot scale to the size of the graphs proposed by Lim et al. (2021). In contrast, for our benchmark, we purposefully collect graphs with less than 50K nodes, allowing us to compare many models for learning under heterophily proposed in the literature.

Another difference is in the domains from which the datasets come. Graphs are a natural way to represent data from different fields; thus, a comprehensive graph benchmark should cover a wide variety of domains. Lim et al. (2021) use social networks (`penn94`, `pokec`, `genius`, `twitch-gamers`), citation networks (`arxiv-year`, `snap-patents`), and a web graph (`wiki`). Our graphs come from other diverse domains and thus naturally complement the benchmark of Lim et al. (2021). Namely, our datasets are a word dependency graph (`roman-empire`), a product co-purchasing network (`amazon-ratings`), a synthetic graph emulating the minesweeper game (`minesweeper`), a crowdsourcing platform worker network (`tolokers`), and a question-answering website interaction network (`questions`).

## D  TWO VERSIONS OF SQUIRREL AND CHAMELEON

There exist two versions of `squirrel` and `chameleon` datasets. One is available on the website of the authors of Rozemberczki et al. (2021),[3] while the other is available on SNAP Datasets.[4] These datasets differ in their edge sets. Pei et al. (2020) adopted the version from SNAP Datasets for their experiments, and thus this version became standard in the literature. In our work, we also use this version, and our observations regarding edges of duplicate nodes only apply to this version. However, regression targets in both versions of the datasets are the same (up to a logarithmic transform). Thus, duplicated targets are present in both versions of the datasets.

---

[3] https://graphmining.ai/datasets/ptg/wiki/
[4] http://snap.stanford.edu/data/wikipedia-article-networks.html

