# OpenReview forum: "A critical look at the evaluation of GNNs under heterophily: Are we really making progress?"
_ICLR.cc/2023/Conference — ICLR 2023 poster_

### Official Review · Reviewer_uqad · 2022-10-18

**Confidence:** 4
**Correctness:** 4
**Technical Novelty And Significance:** 2
**Empirical Novelty And Significance:** 3
**Recommendation:** 5

**Clarity, Quality, Novelty And Reproducibility:**

The authors investigate the recent progress on the heterophily issue. The authors find that the widely used benchmarks have serious drawbacks. Based on the newly developed datasets, the authors point out that standard GNNs achieve competitive
results on heterophilous graphs outperforming most of the specialized models. Overall, this paper is well-written and easy to follow. However, there is no new algorithm. Moreover, it seems that the new data are not publicly available.

**Strength And Weaknesses:**

The strong points of this paper include:
1) This paper is well-motivated and clearly written. The authors examine the datasets and existing methods for heterophily.
2) Five new datasets are developed for heterophily.
3) The evaluation is comprehensive.
The weaknesses include:
1) This paper doesn't provide any new algorithm. The authors just evaluate existing models on their data.
2) The baselines should include some recent works that work on both types of data. For example, JacobiConv, GBK-GNN.

**Summary Of The Paper:**

This paper examines the existing heterophilous data and finds out that they exist some issues. Afterwards, the authors propose several new data. Evaluations show that standard GNNs achieve competitive results outperforming most of the specialized models on heterophilous graphs.

**Summary Of The Review:**

My concern about this paper is that there is no new algorithm. The main contribution of this paper lies in the new datasets. The authors point out the drawbacks of current data, which could be meaningful for the community.

---

> ### Author Response · Authors · 2022-11-08
> **Response to Reviewer uqad**
>
> Thank you for the comments!
>
> Indeed, the focus of our work is not developing a new algorithm but rather demonstrating issues with the commonly used evaluation setting for heterophily-specific models, collecting new datasets, and re-evaluating the models in a more suitable setting. We believe that this provides a valuable contribution to the community.
>
> We plan to add the results for GBK-GNN and JacobiConv models to the revised version of our paper. We will also make our datasets available together with the code for reproducing our results.

---

> ### Author Response · Authors · 2022-11-16
> **Revised paper**
>
> We have uploaded a revised version of our paper, where we have made several improvements. In particular, we have added GBK-GNN and JacobiConv models to our experiments. While these models achieve strong results on some datasets, they are still typically outperformed by standard GNNs such as GAT; thus our conclusions remain the same.
>
> We have also made other improvements to the paper; please see our general response.

---

### Official Review · Reviewer_onVz · 2022-10-25

**Confidence:** 4
**Correctness:** 3
**Technical Novelty And Significance:** 3
**Empirical Novelty And Significance:** 3
**Recommendation:** 6

**Clarity, Quality, Novelty And Reproducibility:**

Clarity: This paper is well-organized and easy to follow.

Quality: This paper is of good quality with detailed discussion and comprehensive experiments.

Novelty: Studying GNNs under heterophily from the perspective of datasets is novel.

Reproducibility: The constructed new benchmark graphs are not provided so the reproducibility could be a question.

**Strength And Weaknesses:**

Strengths:
- Research on Graph neural networks that are able to capture heterophily is an important research problem. This paper studies this problem from a different and novel perspective, i.e., analyzing the issues in these widely used datasets.
- Based on issues of some widely used datasets for heterophilous GNNs, several new benchmarks have been proposed.

Weakness:
- What is the evaluation metric used in Table 2, Accuracy or ROC AUC?
- When introducing Minesweeper and Questions datasets, what's the meaning of additional binary features? Will this feature influence the quality of the dataset, i.e., the correlation between such features and labels can be guaranteed?
- The author mentioned the possible issue in Cornell, Texas, and Wisconsin datasets, i.e., imbalanced distribution. I suggest adding some empirical analysis to better illustrate the impact of such imbalance on previous methods similar to Section 3.1.
- Although the authors claimed an existing benchmark [1] is different from this one ([1] is designed for scalability), more discussions on the differences can be provided from the perspective of data construction.

[1] Large scale learning on non-homophilous graphs: New benchmarks and strong simple methods

**Summary Of The Paper:**

This paper studies the problem of GNNs under heterophily. Different from previous methods, this paper investigates the issues from the dataset perspective. The authors analyzed the issues in five widely used heterophilous datasets and provided some empirical results to illustrate these issues. To solve this problem, a list of new benchmark graphs from different domains as been constructed for evaluating GNNs under heterophily.

**Summary Of The Review:**

This paper studies an interesting and important problem, i.e., GNNs under heterophily. Different from previous methods, this paper investigates the issues from the dataset perspective which is novel and interesting. Although benchmark datasets for similar tasks have been proposed, this paper shows some differences from the existing datasets. Thus, I will recommend a weak acceptance of the paper.

---

> ### Author Response · Authors · 2022-11-08
> **Response to Reviewer onVz**
>
> Thank you for your feedback and comments! We address your concerns below.
>
> > What is the evaluation metric used in Table 2, Accuracy or ROC AUC?
>
> In Table 2 we report accuracy. We write about it in “Training details” on page 8 and will also specify it in the caption of Table 2.
>
> > When introducing Minesweeper and Questions datasets, what's the meaning of additional binary features? Will this feature influence the quality of the dataset, i.e., the correlation between such features and labels can be guaranteed?
>
> The additional binary feature in the Minesweeper and Questions datasets is used to indicate nodes that do not have other features. We use this feature to avoid feeding a neural network with a vector of all zeros, which may negatively affect performance. Using such binary feature followed by a linear layer is equivalent to using a learnable embedding for nodes that do not have other node features.
>
> > The author mentioned the possible issue in Cornell, Texas, and Wisconsin datasets, i.e., imbalanced distribution. I suggest adding some empirical analysis to better illustrate the impact of such imbalance on previous methods similar to Section 3.1.
>
> We will add a table illustrating the class balance in the Cornell, Texas, and Wisconsin datasets to the revised version of our paper. As we note in the paper, there are some extreme issues, such as Texas having a class with just one node. Designing an experiment showing the empirical effect of this imbalance is challenging since we cannot modify these datasets to make them balanced.
>
> > Although the authors claimed an existing benchmark [1] is different from this one ([1] is designed for scalability), more discussions on the differences can be provided from the perspective of data construction.
>
> Our benchmark and the one proposed by Lim et al. (2021) are complementary. The main difference is the size of the datasets: our datasets are smaller and are thus suitable for evaluating the performance of heterophily-specific GNNs, many of which are not scalable. Additionally, our graphs come from different domains and are more diverse than those proposed by Lim et al. (2021). We will add a detailed comparison of the benchmarks to the revised version of our paper.
>
> > The constructed new benchmark graphs are not provided so the reproducibility could be a question.
>
> We will make our datasets available to the community. Additionally, we will provide the code to reproduce our results.

---

> ### Author Response · Authors · 2022-11-16
> **Revised paper**
>
> We have uploaded a revised version of our paper, where we have made several improvements. In particular, we have added a table illustrating the class balance in the Cornell, Texas, and Wisconsin datasets to Appendix B, and we have added a comparison to the benchmark proposed by Lim et al. (2021) to Appendix C. We have also made other improvements to the paper; please see our general response.

---

### Official Review · Reviewer_1RXw · 2022-10-25

**Confidence:** 5
**Correctness:** 3
**Technical Novelty And Significance:** 2
**Empirical Novelty And Significance:** 3
**Recommendation:** 6

**Clarity, Quality, Novelty And Reproducibility:**

The paper is easy to follow, but there are several things that need clarification:
1. How connected is the filtered graph?
2. Do all the duplicate node also have the same label?
3. What fraction of the duplicates are present in train, val and test? If you take the original graph and run any of the models, then split the test set into duplicate nodes and non-duplicate nodes, and measure performance independently, what is the performance split that you obtain? The paper claims memorization as the possible cause of degradation, it will be good to verify this.
4. Is the filtered graph directed or undirected? If directed, is it converted into undirected graph before using in these models? These models cannot handle directed graphs and making the graph symmetric using $A + A^T$ helps.
5. How is the train/val/test created for the filtered graph? Do you create a fresh train, val, test split or simply remove the duplicate nodes from the previous train, val, test split?
6. Are all the baseline models tuned for filtered graph or the previous parameters are utilized for measure the filtered graph performance? These models have sensitivity to hyper-parameter tuning, thus it is important to understand.
7. Are there duplicates in the new dataset? If yes, what fraction?
8. The wiki-cooc and minesweeper dataset already reports pretty high numbers for baseline model like GCN. Is there any way to quantify value of these datasets. For example, an upper bound on performance that might be achieved on these datasets? That will help understand the value of these datasets. Same can be done even for other datasets for uniformity.

If the above questions are clarified and If the reported numbers are properly verified and double checked, then the paper shows a very interesting finding and proposes several new interesting datasets on which newer model that can address heterophily can be built. This will be of great help to community.

**Strength And Weaknesses:**

# Strengths:
1. The paper is clearly written and easy to follow.
2. The raised issue of duplicates is very interesting.
3. The reported results significant gap in performance.
4. The new datasets presented show wide variety in domain, size, heterophily and class balance.
***
# Weaknesses:
1. There is no reported statistics on the filtered graphs.
2. GPRGNN can be reduced to SGCN model [A]. Typically SGCN model is quite close in performance to GCN. Also, note that for homophilic datasets, GPRGNN results are on par with GCN. Additionally, GPRGNN was also evaluated on synthetic benchmarks built from stochastic block models which are not likely to have the duplicate issue. It showed good performance even on those. All of this suggests that GPRGNN model should be at least as competitive as GCN on any dataset. So, it is quite odd that GCN gets 40.70% on filtered squirrel whereas GPRGNN is at 33.57%. Similar issues are present in results reported for the new datasets. This makes the reported numbers less believable.
3. The paper does not report whether there are duplicate issues in the new datasets.
4. The wiki-cooc dataset already report pretty high numbers. Despite having low homophily, GCN numbers are already pretty high suggesting quite possibly strong node features. So, the value of the dataset is somewhat unclear. Same with minesweeper dataset.
***
# References:
[A] Simplifying Graph Convolutional Networks, Felix Wu, Tianyi Zhang, Amauri Holanda de Souza Jr., Christopher Fifty, Tao Yu, Kilian Q. Weinberger, ICML 2019.

**Summary Of The Paper:**

Studying heterophily of graphs and their potential impact of learning GNN models is an important area of research. However, the datasets currently used for evaluating the various models could have potential biases and issues. This paper identifies one such issues with these datasets and finds that mitigating the issue causes several SoTA models for heterophily to do poorly. The paper then goes ahead and creates new heterophilic datasets which have wider variety in terms of domains, sizes and heterophily. The paper shows that simple modifications of GAT architecture on these datasets work better than most of the SoTA models for heterophily.

**Summary Of The Review:**

Overall, the paper two important points:

1. The current heterophilic datasets have issue of duplication. And that current research using these dataset is flawed because of it. However, the numbers reported for various models are not quite believable. And it would be worthwhile to double check and verify the correctness of these numbers. Also, some more analysis on whether duplicates are present in large proportions in train and test and not only in large proportions but possibly skewed proportions. Any other analysis, in addition, would be helpful. If all of this leads to the claim that memorization is truly the cause of drop in performance, then the numbers might be more believable.

2. The paper presents 5 new datasets. However, for two of these datasets, the numbers for base models like GCN are already pretty high. So, the value of these datasets for heterophilic graph neural network research is unclear.

Based on these, I am inclined to mark this paper as marginally below acceptance threshold.

---

> ### Author Response · Authors · 2022-11-08
> **Response to Reviewer 1RXw**
>
> Thank you for your constructive feedback and suggestions for improving the paper! We reply to your comments and suggestions below; the numbering corresponds to the comments in “Clarity, Quality, Novelty And Reproducibility”.
>
> 1. The filtered graphs are connected. As described in Section 3.1, for almost all groups of duplicates, there exists a node in the dataset with the same outgoing edges that the duplicates in this group have but with additional incoming edges. Since these nodes remain in the graph after removing the duplicates, the graph's connectivity is not lost.
> 2. All duplicates in one group have the same label (as we write in Section 3.1, they even have the same monthly traffic on which the labels are based), but there are many groups from different classes. We will add a table showing the distribution of duplicates across classes in the revised version of our paper.
> 3. Thank you for the suggested experiment! We will run it and provide its results in the revised version of our paper.
> 4. We treat these graphs (both before and after filtering) as undirected, as is commonly done in the literature.
> 5. We do not create new train/val/test splits but rather use the original splits with duplicate nodes removed (duplicates exist in all splits).
> 6. Indeed, many tested models are sensitive to the choice of hyperparameters. Therefore, we tune hyperparameters on each dataset, including separate tuning on the filtered datasets.
> 7. We suppose that a large number of duplicates in the squirrel and chameleon datasets is a result of some peculiarity of the data-gathering process used for collecting the original datasets. Thus we do not generally expect to find duplicates in other graphs. We checked our graphs for duplicates and did not find anything suspicious. In minesweeper, roman-empire, and wiki-cooc, all neighborhoods are unique. In questions, there are degree one nodes with the same neighbor - it is not unusual for a social network. In amazon-ratings, there are a few cases when the neighborhoods are the same, but the neighborhood size is small.
> 8. We are unaware of a method that would allow one to find the upper bound for the performance on such complex datasets. However, while some models achieve strong results on wiki-cooc and minesweeper datasets, others show significantly weaker results (especially on minesweeper). Thus, these datasets can help differentiate these models, and we believe these datasets can be useful for determining which models work well on heterophilous graphs.
>
> Regarding your concern about the weak performance of GPR-GNN: we use the original implementation of this model provided by the authors, and to ensure the reproducibility of our results, we will make our code available.

---

> > ### Comment · Reviewer_1RXw · 2022-11-25
> > **Thanks for the response!**
> >
> > Follow up comments:
> > 1. That is good to know.
> > 2. It would be great, if you can add this.
> > 3. Thanks for running this experiment, it would be good to have these results in the paper.
> > 4. +1
> > 5. Do you believe that the train/val/test splits are biased by the removal of duplicate nodes? For example, what if all duplicate nodes were in train set or val set or test set etc. then such a thing could bias the final result. Can you add some explanation as to what could be happening here?
> > 6. That is good to know.
> > 7. It might be good to add details of the duplicate nodes in the proposed dataset in the paper. As the reduced numbers could add to the proposed analysis.
> > 8. One way to do that could be to randomly remove some percentage of bad edges (edges connected to nodes with different labels) and running a model like GCN. This percentage could be varied and performance gains can be reported to show how much improvement might be possible if we build a good heterophily model on these datasets. There could be other ways to demonstrate this as well, but this is just one way.

---

> > > ### Author Response · Authors · 2022-11-26
> > > **Some clarifications**
> > >
> > > Thanks for your reply!
> > >
> > > During the discussion stage, we have uploaded the revised version of the paper in which we have added the experiments we promised. Please see the current version of the paper and our [summary of changes](https://openreview.net/forum?id=tJbbQfw-5wv&noteId=MLuGi45msi). Below are some additional clarifications.
> > >
> > > > 2. It would be great, if you can add this.
> > >
> > > We have added a table showing the distribution of duplicates across classes; please see Table 5 on page 13.
> > >
> > > > 3. Thanks for running this experiment, it would be good to have these results in the paper.
> > >
> > > We have added the results of this experiment for GraphSAGE; please see Table 1 on page 4. As expected, GraphSAGE achieves much better accuracy on duplicates than on non-duplicates, confirming that the model relies on the train-test data leakage to make predictions for duplicates.
> > >
> > > > 4. +1
> > >
> > > We have a comment about this in Appendix A, second sentence.
> > >
> > > > 5. Do you believe that the train/val/test splits are biased by the removal of duplicate nodes? For example, what if all duplicate nodes were in train set or val set or test set etc. then such a thing could bias the final result. Can you add some explanation as to what could be happening here?
> > >
> > > Since splits are random, the duplicates are distributed uniformly across them. We checked that the train/val/test proportions are similar for the original and filtered datasets. We believe that the difference in performance for the original and filtered datasets is caused by data leakage, as we discuss in Section 3.1.
> > >
> > > > 7. It might be good to add details of the duplicate nodes in the proposed dataset in the paper. As the reduced numbers could add to the proposed analysis.
> > >
> > > We will add these details to the next revision (at this moment, new revisions cannot be made).
> > >
> > > > 8. One way to do that could be to randomly remove some percentage of bad edges (edges connected to nodes with different labels) and running a model like GCN. This percentage could be varied and performance gains can be reported to show how much improvement might be possible if we build a good heterophily model on these datasets. There could be other ways to demonstrate this as well, but this is just one way.
> > >
> > > This is an interesting additional analysis of the datasets; thanks for the suggestion. However, it cannot be used as a reliable upper bound for the original dataset. Note that if heterophilous neighbors are informative, sparsifying the dataset may reduce the model performance if this model is able to use this information. For example, in the minesweeper dataset, the nodes by design provide helpful information about their neighbors regardless of the class. Thus, on this dataset, any changes to the graph structure will probably lead to decreased performance.
> > > If requested by the reviewers, we are open to adding experiments of this type to the final version of the paper. We also believe that adding one more dataset (workers) addresses the original problem: for this dataset, there is a large room for possible improvements.
> > >
> > > We will be happy to hear your thoughts about the revised paper and are open to further discussions.

---

> > > > ### Comment · Reviewer_1RXw · 2022-12-05
> > > > **Thanks for the response**
> > > >
> > > > About point 8 above, the point is not to get an upper bound, it is just to see that there is potential for improvements that are possible with these datasets.
> > > >
> > > > I thank the authors for all the response and addressing the concerns. I have adjusted my score accordingly.

---

> ### Author Response · Authors · 2022-11-16
> **Revised paper**
>
> We have uploaded a revised version of our paper, where we have made improvements based on your suggestions. In particular, we have added an experiment where for the squirrel and chameleon datasets, we measure the performance on duplicates and non-duplicates separately. The results are provided in Table 1. As expected, the performance is much better on duplicated nodes, which confirms that the model relies on the train-test data leakage to make predictions.
>
> We have also made other improvements to the paper; please see our general response.

---

### Official Review · Reviewer_ePQD · 2022-10-25

**Confidence:** 3
**Clarity, Quality, Novelty And Reproducibility:** Please see Strength And Weaknesses
**Correctness:** 4
**Technical Novelty And Significance:** 3
**Empirical Novelty And Significance:** 3
**Recommendation:** 8

**Strength And Weaknesses:**

Pros:

1. This paper comprehensively summarises the related work on GNNs under heterophily.
2. This paper analyses the drawbacks of the widely used benchmark (Pei et al. 2020) and performs experiments to validate the effect of removing duplicate nodes.
3. The novelty of the proposed new datasets is reasonable to me (using different homophily measure; from diverse domains).

Cons:

Regarding another recent non-homophilous graph benchmark (Lim et al. 2021), this paper only mentions its scale. I expect a more detailed comparison with it (e.g., what are the characteristics of those datasets? Are they suitable for evaluation under heterophily?). This may help people differentiate them and make them complement each other.

**Summary Of The Paper:**

This paper proposes a new benchmark for evaluating GNNs under heterophily. Adjusted homophily and label informativeness are used to characterize new datasets from diverse domains. Extensive experiments are conducted to evaluate various GNN methods on these new datasets, and results show that standard GNNs outperform heterophily-specific GNNs.

**Summary Of The Review:**

This paper will be of interest to the non-homophilous GNN community

---

> ### Author Response · Authors · 2022-11-08
> **Response to Reviewer ePQD**
>
> Thank you for your comments and positive feedback!
>
> Indeed, our benchmark and the benchmark proposed by Lim et al. (2021) complement each other. Besides being smaller and thus suitable for evaluating heterophily-specific GNNs, many of which are not scalable, our graphs come from different domains and are more diverse than the ones proposed by Lim et al. (2021). We will add a detailed comparison of the benchmarks to the revised version of our paper.

---

> ### Author Response · Authors · 2022-11-16
> **Revised paper**
>
> We have uploaded a revised version of our paper where we have added a comparison to the benchmark proposed by Lim et al. (2021); see Appendix C. We have also made other improvements to the paper; please see our general response.

---

> ### Comment · Reviewer_ePQD · 2022-11-21
> **Review Update**
>
> Thanks to the authors for the detailed response. I have also read other reviewers' comments. My concerns have been addressed and I have raised my score.

---

### Author Response · Authors · 2022-11-16
**Revised paper: summary of changes**

We have uploaded a revised version of our paper, where we have made improvements based on the reviewers’ comments. The main changes are as follows:
1. We have rewritten Section 3.1 to emphasize that duplicates in the squirrel and chameleon datasets create a train-test data leakage which affects the performance of the models on these datasets. As suggested by Reviewer 1RXw, we confirm that the models utilize this leakage by comparing the performance on duplicates and non-duplicates, finding that the performance on duplicated nodes is much better. We report these numbers in Table 1. Removing data leakage strongly changes the relative performance of the models. For example, FSGNN is the best model by a significant margin on both original datasets, however, on filtered squirrel and chameleon it achieves only 12th and 13th places respectively. To illustrate the relative performance change, we have added model ranks on original and filtered datasets to Table 2.
2. To further increase the coverage and diversity of the benchmark, we added one more dataset. This dataset - workers - is based on data from a crowdsourcing platform, and the task is to predict which workers were banned. On this dataset, the best results are achieved by standard GNNs (GAT and GT), which is in line with our results on other datasets.
3. As suggested by Reviewer uqad, we have added two recently proposed specialized GNNs to our experiments: GBK-GNN and JacobiConv. While these models achieve strong results on some datasets, they are still typically outperformed by standard GNNs such as GAT. Thus, our conclusions remain the same.
4. We have added a more detailed comparison of our datasets with those proposed by Lim et al. (2021); see Appendix C.
5. We have made the datasets and code publicly available; see Section 5.1.

If there are any additional suggestions or questions, we will be happy to address them.

Sincerely, Authors

---

> ### Author Response · Authors · 2022-11-19
> **Approaching the end of Discussion Stage 1**
>
> Dear Reviewers,
>
> During stage 1 of the discussion period, we did our best to address all of the concerns. We will be glad to hear your thoughts about the revised paper. We are open to additional questions and discussions.
>
> Sincerely, Authors

---

### Decision · Program_Chairs · 2023-01-20

**Decision:**

Accept: poster

**Justification For Why Not Higher Score:**

While this method would have potential for a spotlight, the authors would have probably needed to conduct a significantly more thorough study of the effects they discovered in order to warrant it, in my opinion.

**Justification For Why Not Lower Score:**

My opinion, concurred by other reviewers, is that this work passes the bar for ICLR, and will be valuable to several groups within graph representation learning. I am absolutely certain about this assessment.

**Metareview: Summary, Strengths And Weaknesses:**

The authors of this paper reveal an unexpected point about existing heterophilous datasets---that one of the key points making them challenging for classical GNNs may be issues such as duplicated nodes in the data. They follow up by testing their method on "cleaned up" versions of these benchmarks, and including additional tasks with comprehensive evaluation. All reviewers agreed this method is valuable, and after thoroughly discussing the pros and cons with them, I am confident that this paper should be accepted. More details may be found in the AC-Reviewer meeting summary.

**Note From Pc:**

if the above contains the word "oral" or "spotlight" please see: "oral" presentation means -> notable-top-5% and "spotlight" means -> notable-top-25%. As stated in our emails, we are disassociating presentation type from AC recommendations

**Summary Of Ac-Reviewer Meeting:**

The reviewers highlighted the following reasons about accepting the work:

* It's a comprehensive evaluation of existing GNN proposals under heterophilous constraints, a very important open problem in the area.
* It detects serious drawbacks of previous benchmarks.
* The proposed benchmark has significant differences to existing ones---new measures of homophily, and a diverse basis of tasks.
* Creating new datasets, and establishing new best practices, is generally perceived as highly important.
* The authors adequately addressed the concerns of the reviewers in the rebuttal.

The key concerns were:

* It would be nice to include a more comprehensive set of baselines;
* It would have been useful to have more detailed qualitative or quantitative experiments around the effects of node duplication;
* Some reviewers raised the issue that the reported evaluation numbers seemed a little unusual, and they were unsuccessful at reproducing them so far;
* There doesn't seem to be a new method proposed---the contribution is mainly in the data.

My opinion, which the reviewers concurred with in the call, is that the pros significantly outweigh the cons. So long as the authors make their full source code publicly available (which they pledged to do), anyone could add more baselines, and efforts to reproduce the results can continue. Further, I do not agree that a successful paper needs to propose a new method, _or_ conduct detailed qualitative studies---if it points out something that the community should find interesting, and does it well, it is admissible as an ICLR paper. And in this case, the authors have certainly done a good job bringing a new and interesting observation to the community. More work can always be done, but it's not the responsibility of the authors to do all of it in one paper.